

# Does pruning affect the structural and ecological productivity of Juniper woodlands in the eastern Hindu Kush?

Nasir Ud Din[1], Nasrullah Khan[1], Rafi Ullah[2], Mohammad K. Okla[3], Mostafa A. Abdel-Maksoud[3], Ibrahim A. Saleh[4], Hashem A. Abu-Harirah[5] and Tareq Nayef AlRamadneh[5]

[1] Department of Botany, University of Malakand, Khyber Pakhtunkhwa, Pakistan
[2] Department of Botany, Dr. Khan Shaheed Govt: Degree College Kabal Swat, Khyber Pakhtunkhwa, Pakistan
[3] Department of Botany and Microbiology, College of Science, King Saud University, Riyadh, Saudi Arabia
[4] Faculty of Sciences, Zarqa Universty, Zarqa, Jordan
[5] Department of Medical Laboratory Sciences, Faculty of Allied Medical Sciences, Zarqa University, Zarqa, Jordan

Corresponding authors
Nasir Ud Din,
nasir.akash88@yahoo.com
Nasrullah Khan,
nasrullah@uom.edu.pk

## ABSTRACT

*Juniperus* species play a crucial role in the ecological dynamics of the eastern Hindu Kush region in Pakistan, growing either as mono-specific stands or co-occurring in species-poor communities on mountainous scree slopes, as well as near agricultural and residential areas. Despite their limited population, these trees offer a diverse range of ecosystem services, emphasizing the intricate interdependence between human and natural ecosystems. Given their ecological and commercial importance, local people employ silvicultural practices, such as pruning, to ensure sustainable management. The present study, therefore, aimed to assess conventional pruning practices in the region to understand their impacts on stand structure and ecological productivity. The findings indicated that pruning at an intensity of 30–40% was particularly effective, significantly enhancing dendrometric and reproductive traits, such as height, growth rates, and the production of viable seeds (ANOVA, $p < 0.05$). Additionally, both diameter at breast height (DBH) and diameter at root collar (DRC) showed positive trends, although the effects were not statistically significant (ANOVA, $p > 0.05$). Conversely, pruning intensities exceeding 60% exhibited adverse effects on the tree metrics. Our results also highlight the importance of pruning intensities in regulating understory vegetation, soil nutrient dynamics, and the carbon storage capacity of junipers. Notably, moderate pruning demonstrates positive impacts on living carbon biomass (LCB) and on soil organic carbon (SOC) density. In conclusion, adopting moderate-intensity pruning techniques with standard scientific measures could be an effective strategy, not only for improving the structural parameters and carbon budgets amid changing climate conditions but also for ensuring long-term economic advantages in the region.

## INTRODUCTION

Different silvicultural techniques have been employed for sustainable forest management and to enhance ecological productivity (*Clark & Matheny, 2010*). Among these techniques, pruning is an important practice that plays a pivotal role in regulating dendrometric responses through activities such as cutting, thinning, or treating different plant parts, including branches, shoots, and occasionally leaves (*Clark & Matheny, 2010*; *Zhang et al., 2018*). Pruning can be categorized on the basis of season, *i.e.*, spring, summer, or autumn pruning and according to pruning intensities, which may range from light to heavy pruning (*Zhang et al., 2018*). Additionally, winter pruning of various grapevines is a common practice observed across the globe (*Intrieri & Poni, 1997*). Research has indicated that pruning is usually used to improve the structural shape of trees during the establishment period, aiming to produce knot-free timber wood (*Neilsen & Pinkard, 2003*; *Maurin & DesRochers, 2013*). Pruning can induce stress in tree growth or trigger growth rates by improving secondary growth (*Springmann, Rogers & Spiecker, 2011*). It is described that pruning may enhance the quality of photosynthesis in the remaining foliage (*Medhurst et al., 2006*), while some studies also suggest its negative effects (*Turnbull, Adams & Warren, 2007*). In a study involving *Diospyros melanoxylon* Roxb, pruning demonstrated significant impacts, with pruned plants exhibiting a five-fold increase in healthy leaf production compared to the non-pruned population of the same species (*Mehta, Jain & Rajkumar, 2020*). Furthermore, phenols and carbohydrates contents were also notably elevated in the pruned plants (*Mehta, Jain & Rajkumar, 2020*).

According to *Torres, Luján & Pineda (1995)*, pruning is considered a cleaning practice rather than a silvicultural technique. While there are few recommendations regarding pruning intensities for forest plantations, but a general suggestion is that pruning range, up to 50% may be a suitable after first thinning practices (*Keogh, 1987*). Moreover, pruning techniques have been used for multiple purposes *e.g.*, in young plants, pruning is primarily aimed at supporting fruit load (*Vossen & Devarenne, 2007*; *Gregoriou, 2009*; *Therios, 2009*). The pruning is sometimes applied for pest and disease control (*Tombesi & Tombesi, 2007*) and is also used on some high-value tree species to enhance stem quality, vigor, and overall tree development (*Medhurst et al., 2006*; *Pinkard & Beadle, 1998*). Pruning is crucial for enhancing light penetration, potentially stimulating the development of flower buds and subsequent fruit sets in fruit-bearing plants (*Marini, Sherif & Smith, 2020*). Furthermore, pruning remains an important silvicultural technique aims to mitigate forest fires by establishing adequate spatial separation between surface fuels and the live canopy (*Hevia et al., 2018*).

*Juniperus* L. is a genus within the Cupressaceae family, encompassing 70 species with a widespread distribution globally, ranging from lowlands to the tree-line (*Adams, 2008*). In Pakistan, different species of *Juniperus* have been reported in mountainous regions including Chitral district (*Stewart, Nasir & Ali, 1972*). Junipers are well known for their drought resistance, and adaptability to harsh climatic conditions and nutrient-poor soil (*Sarangzai et al., 2012*). In the study area, four species have been identified flourishing between the elevations of 2,000–3,500 m (*Din, 2018*). These species are found

in mountainous habitats, and along with human settlement areas, including private forest patches, and alongside the residential and agricultural lands. In this context, local inhabitants employ various silvicultural techniques, including thinning and pruning without knowing its broader implications of pruning on ecological systems. Communities engage in pruning to fulfill various needs, such as obtaining knot-free timber wood, ensuring sufficient light for underlying agricultural land, and procuring fuel-wood. Pruning is known to influence tree growth, fruit quality, and overall productivity, potentially enhancing the vigor and resilience of the forest. However, the effects of these practices on *Juniperus* species within natural habitats remain largely understudied. Therefore, this study hypothesized that pruning has beneficial impacts on the juniper forests. To test this hypothesis, we evaluated differences in key plant functional traits between pruned and non pruned sites, assuming that any positive effects of pruning will be reflected in these measurable traits. Consequently, the aims of our study were focused on addressing the following questions; (i) how do conventional pruning techniques affect the forest structures and ecological productivity? (ii) to what extent can non-scientific or the lack of standardized pruning practice cause harm, and (iii) does pruning play an effective role in climate change mitigation? If so, which level of pruning intensity proves to be effective? Achieving these aims will provide critical baseline data on how pruning improves and sustains juniper woodlands, offering insights into its potential to enhance forest sustainability, support human livelihoods, and contribute to ecosystem services.

## MATERIALS AND METHODS

### Study area

Chitral, previously the largest district of Khyber Pakhtunkhwa with a total area of 14,850 km$^2$, later underwent administrative division resulting in Lower and Upper Chitral districts. Chitral geographically extends between latitudes 35°15′06″ to 36°55′32″N and longitudes 71°11′32″ to 73°51′34 E, in the northern part of the province (Fig. 1). The area lies at elevation ranges of 1,100 m to 7,708 m above sea level (*Khan, Ahmed & Shaukat, 2013*). The climate data (2016–21) show that Chitral is characterized by arid conditions, with the majority of rainfall occurring between January and April, accounting for 73% of the total rainfall and representing the wettest months of the year. In contrast, the period from June to September experiences the lowest rainfall, accounting for only 3.68% of the total, making these months the driest of the year. June, July, and August are also the hottest months, with temperatures ranging from 34–36 °C, while December, January, and February are the coldest months in the area. The relative humidity (RH) maintains an average of 72.1% at 0300Z, with highest recorded RH in August, September, and October at 80.5, 83.7, and 84.8%, respectively (Fig. 2).

The forest of the area is dominated by conifers, particularly *Cedrus deodara*, *Juniperus excelsa*, *Pinus geradiana*, *Pinus wallichiana*, *Abies pindrow*, and *Picea smithiana* (*Khan et al., 2022*), followed by broadleaved species like *Quercus incana*, *Quercus dilatata*, *Quercus baloot*, *Juglans regia*, *Tamarix dioica*, and *Betula utilis* (*Khan, Ahmed & Shaukat, 2013*). Our present study focuses on the Mastuj Valley in Upper Chitral, located at approximately

 

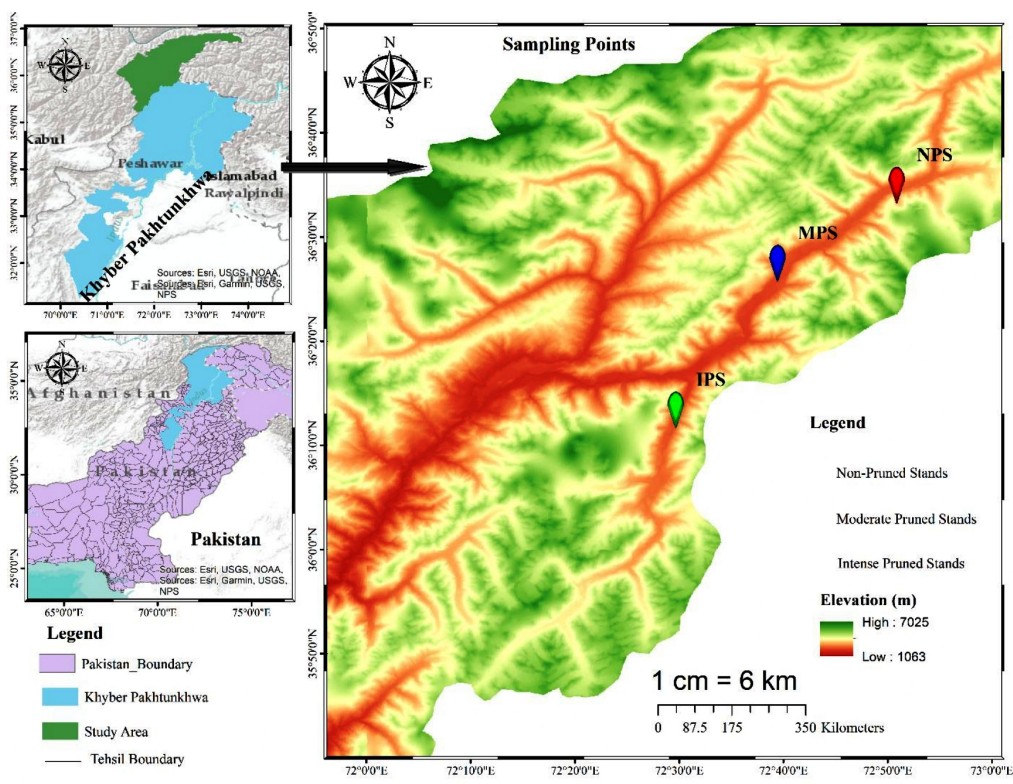

**Figure 1** Map extracted from Arc-GIS, showing distribution of moderate, intensive, and non-pruned juniper forest stands.

36.3°N latitude and 72.5°E longitude. Geographically, it is bordered by Gilgit to the east, Afghanistan to the north, and District Swat to the east and south (Fig. 1). In the studied valley, *Juniperus semiglobosa*, *Juniperus excelsa*, *Salix alba*, *Salix nigra*, and scattered birch patches dominate the wild habitats.

## Field data acquisition

Data were collected in August 2022 from three representative juniper forest stands located in close proximity based on pruning intensity as the key criterion. The first sampling site represented forested land exhibiting a moderate level of pruning. Inhabitants in this area predominantly engaged in mixed forest management practices, particularly to facilitate grazing activities by utilizing the forest floor. Our second sampling site was situated alongside the agricultural and residential lands and adjacent to orchards. Notably, pruning practices in this setting were more pronounced, driven by the necessity for sunlight exposure essential for cereal crops and understory vegetation. In contrast, the third sampling site, characterized by minimal anthropogenic disturbance, resulting in limited pruning activities, therefore, served as the control group (proxy) (Fig. 3). Vegetation and soil attributes in pruned and non-pruned sites were sampled using 20 × 20 m (400 m$^2$) plots for tree stratum and 5 × 5 m subplots for understory vegetation along 200 m line transects across each stand (*Tesfaye, Gardi & Blaser, 2019*). The entire plants both in the

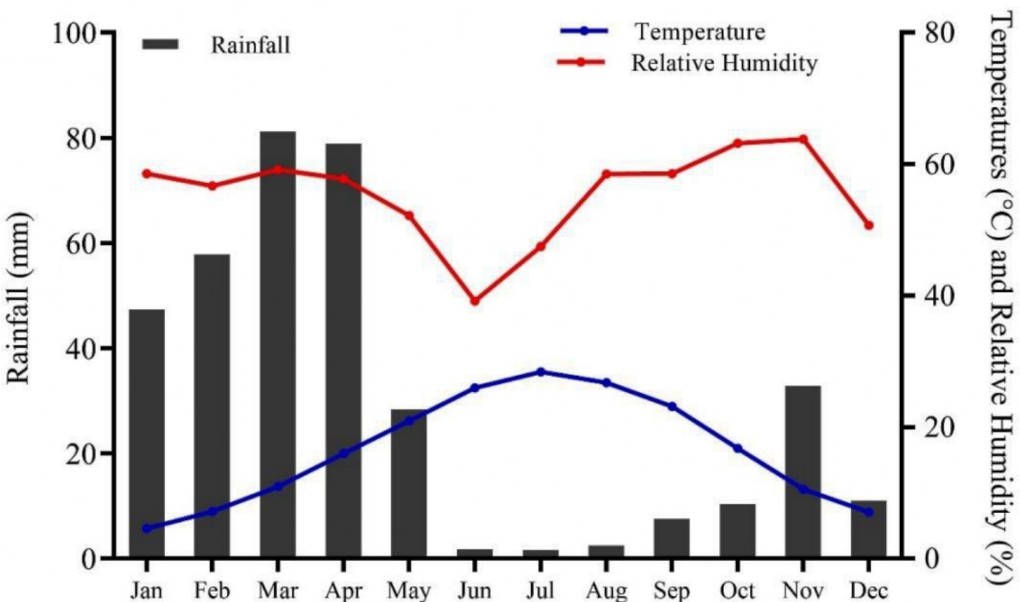

**Figure 2  Climate of Chitral district, spanning from 2016–2021 (Source: Pakistan Meteorological Department, PMD).**

overstory and understory strata were identified with the help of the flora of Pakistan (*Stewart, Nasir & Ali, 1972*), pictorial guides, and were confirmed from Plantworld.com (https://plantworld.com/; Kew Botanical Garden). Furthermore, the density (individuals $ha^{-1}$) of both trees and understory species was calculated, and the Shannon–Weiner diversity (H') index was determined (*Wu et al., 2015*).

Key dendrometric parameters, including tree height, DBH, DRC, and crown base height (CBH) were measured using standard forestry methods (*Bezos et al., 2012*). During the sampling phase, these three sites were classified into three categories; non-pruned stand (hereafter NPS), moderately pruned stand (30–40%; hereafter MPS) and intensive pruned stand ($\geq$ 60%; hereafter IPS), based on the ratio of the crown base height (CBH) to the total tree height (TH) (*Huang et al., 2023*). For categorization of the pruning intensities we used the given equation CBH/TH × 100, for details see in Fig. 4.

For growth rate assessment, more than 40 healthy trees of the targeted species in each forest stand were bored at breast height in damage-free parts through Swedish increment borer (five mm diameter) following *Fritts (1976)* and *Speer (2010)*. The extracted core samples were then stored in plastic straws along with the site description, following standard methodology (*Kahveci, Alan & Köse, 2018*). In the laboratory, wood samples were air-dried and subsequently sanded with different girth-size sandpapers until the visibility of the growth rings (*Fritts, 2001*). To assess age and growth rates, annual growth rings were carefully counted with the help of magnifying lenses and a binocular microscope, following *Ahmed et al. (2011)*. To evaluate reproductive efforts, the numbers of female cones were randomly collected from a total of five trees at 4–5 branches at each stand. Likewise, we

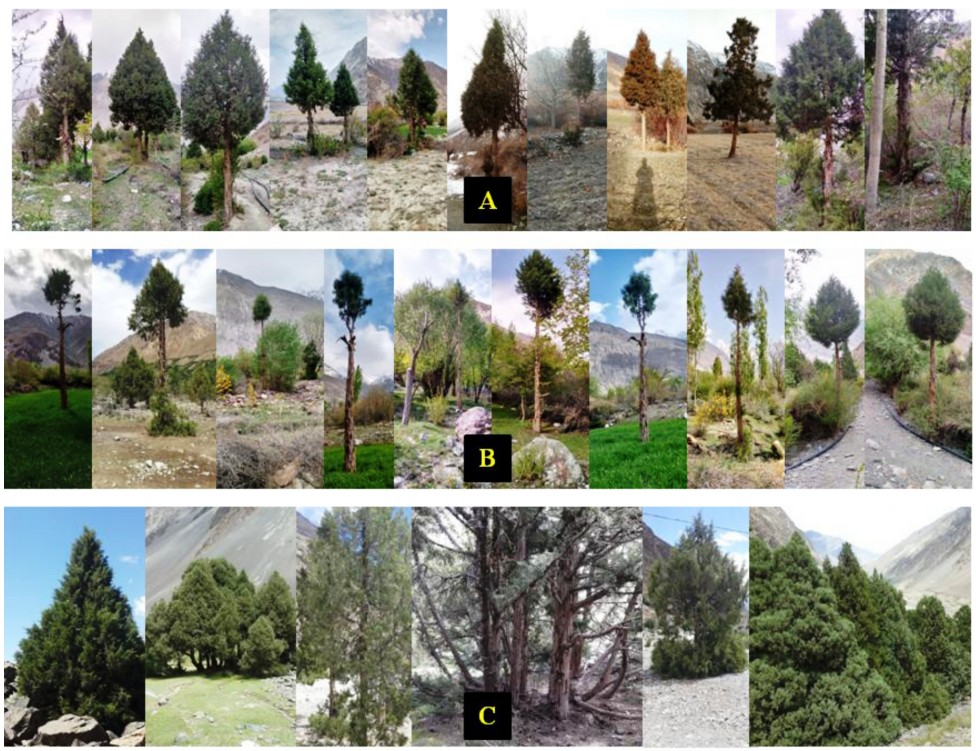

**Figure 3** Different pruning intensities in juniper woodlands: (A) moderate (B) intensive (C) non-pruned.

selected 100 subsamples of cones for counting the filled seeds (living embryos) by cutting it transversely (*Ortiz, Montserrat & Salvador, 2002*). Generally, 3–5 and rarely six seeds are present per female cone of *J. excelsa* and 2–3 seeds in *J. semiglobosa* (*Din, 2018*).

## Environmental and soil variables measurements

The geospatial coordinates including latitude, longitude, and elevation of each site were recorded using a geographical positioning system (GPS). Soil samples (weighing one kg each) were collected in triplicate within each stand with one sample taken from each of the two corners and the center, at various depths (ranging from 0–15 cm and 15–30 cm), followed by *Weil & Brady (2017)*. These samples were combined to create a composite sample which was then air-dried and passed through a two mm sieve to remove any extraneous materials (*Ullah et al., 2022*). Finally we analysed the soil samples for various parameters, such as $CaCO_3$ percentage, soil texture properties (clay, sand, and silt %), pH, and organic matter (OM %), following *Bartels, Bigham & eds (1996)* and *Dane, Topp & eds (2002)*. Additionally, electrical conductivity (EC in dS m$^{-1}$), total phosphorus (mg kg$^{-1}$), and potassium content (mg kg$^{-1}$) were also analysed in the Agricultural Research Station (ARI) Chitral, Khyber Pakhtunkhwa. By using clay and sand percentages of soil, we further calculated soil hydraulic properties (SHP) such as conductivity (mm hr$^{-1}$), saturation (%), bulk density (g cm$^{-3}$), wilting point (kPa), water availability (%), and field capacity (kPa),
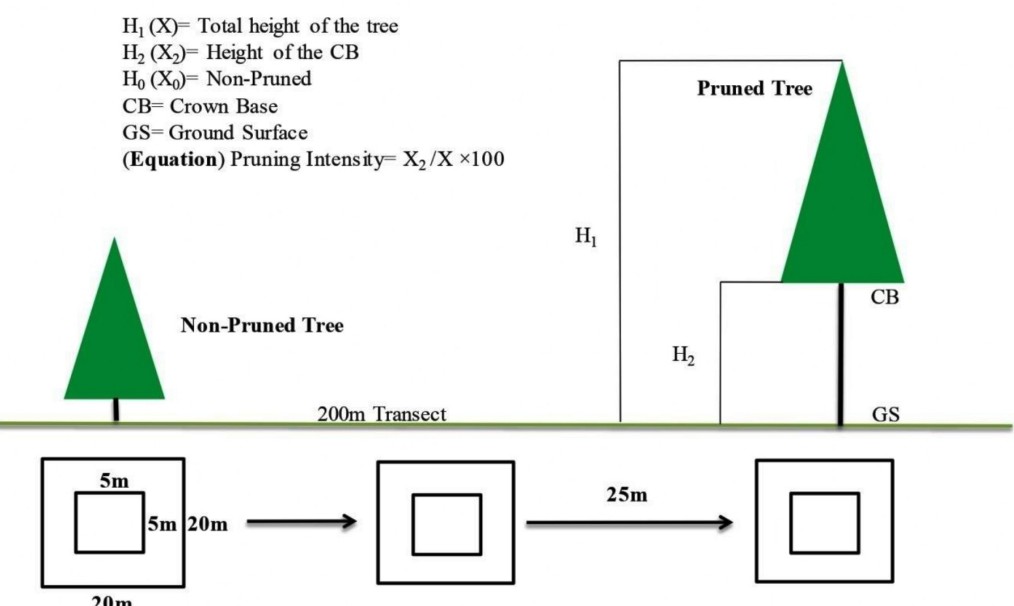

**Figure 4** Experimental layouts, exhibiting sampling procedure for both over and understory strata.

through online calculator (http://www.dynsystem.com/netstorm/soilwater.html) following *Saxton et al. (1986)*.

## Calculation of biomass and carbon stocks

We employed a previously established allometric equation (Eq. (1)), to calculate trees biomass and carbon stock (*Chave et al., 2014*). Predictors such as DBH, height, and species-species wood density (obtained from the Pakistan Forest Institute, Peshawar) were used for biomass calculations (*Gebeyehu et al., 2019*).

$$AGB\ (Kg) = 0.0673\ (pD^2H)^{0.976}. \tag{1}$$

In the equation, AGB represents aboveground biomass, $p$ is the wood density of the species (g cm$^{-3}$), D is the DBH (cm), and H is the height (m). Carbon estimation was performed by assuming 50% of the total biomass following *IPCC (2006)* guidelines, whereas below biomass estimation was derived using a simplified root-to-shoot ratio, *i.e.,* 1:5 (20%) of AGB (*Pearson, 2007*).

## Soil carbon stocks

Soil organic carbon was determined using soil depth (cm), bulk density (g cm$^{-3}$), and SOC %, by using Eq. (2) (*Pearson, 2007*).

$$SOC\ stocks = BD \times d \times SOC\% \times CF \tag{2}$$

where SOC represents soil organic carbon, BD is the bulk density (g cm$^{-3}$), d is soil depth (cm), and CF is the conversion factor for rock fragments. Since CF was not applicable in our case due to the absence of significant rock fragments, therefore, we assumed CF = 1 for its omission

## Calculation of $CO_2$ sequestration

$CO_2$ sequestration was calculated by multiplying the conversion factor (3.67), which represents the ratio of the atomic weights of carbon (12) to carbon dioxide (44), accounting for the additional oxygen atoms in $CO_2$, with TLC (AGBCS, BBCS) using Eq. (3) (*Afzal & Akhtar, 2013*; *Toochi, 2018*).

$$CO_2 \text{ (ton } CO_2) = 3.67 \times TLC \qquad\qquad (3)$$

where $CO_2$ is $CO_2$ sequestration and TLC is total living carbon (Mg C ha$^{-1}$, where 1 Mg = $10^6$ g).

## Statistical analysis

Descriptive statistics such as mean and standard error (mean ± SE) were used to illustrate the mean differences among these three forest stands (*Fadairo et al., 2023*). Differences in the means among the sites were evaluated using a one-way analysis of variance (ANOVA) at a 0.05 significance level (*Chaiya et al., 2023*), whereas to elucidate the linear associations between distinct pruning treatments and selected variables, we used Pearson correlation (r) by following *Bezos et al. (2012)*. Data normality (ANOVA) was assessed using the Shapiro–Wilk test, with a significance level of 0.05, if the *p*-value from the Shapiro–Wilk test was greater than 0.05, the data were considered normally distributed (*Shapiro & Wilk, 1965*). For statistically significant variables, pair wise comparisons were conducted using Tukey's HSD test (*Pearson, 2007*). A 95% confidence interval for the correlation coefficients was computed using Fisher's z- transformation, where the correlation coefficients were transformed to z-scores, and confidence intervals were constructed in the z-scale before back-transforming to the original correlation scale (*Steur et al., 2020*). All statistical analyses were performed using IBM SPSS Statistics version 30.0.0.0 (IBM Corp., Armonk, NY, USA) and OriginPro version 2024b software (OriginLab Corporation, Northampton, MA, USA).

# RESULTS

## Forest characteristics and physical environment

The forest comprises six native tree species from four families, including two species of *Juniperus* (*J. semiglobosa* and *J. excelsa*) in the overall overstory composition. The sites are geographically located at elevations between 2,607 and 2,668 m (a.s.l.), facing northwest (126–153°) and southwest (260°) aspects (Table 1). *J. semiglobosa* is the dominant species in all forest stands, with importance values of 31–36%, followed by *J. excelsa* (IV = 23–29%), except in the intensively pruned site (IPS), where *Elaeagnus angustifolia* co-dominates with 18.5% IV. *Salix alba* contributes 16.45% in the moderately pruned site (MPS), while two other species contribute less than 5%. The highest IV for *Salix alba* (15.21%) was observed in the IPS, with *Hippophae rhamnoides* having the lowest IV across all sites. *Populus alba*, considered a minor species, ranged from 3 to 8.6% IV in the MPS and IPS but was absent in the non-pruned site (NPS). Tree density was highest in the MPS (301 stems ha$^{-1}$), while basal area was greatest in the NPS (26.84 m$^2$ ha$^{-1}$). The IPS had the lowest density (263 stems ha$^{-1}$) and basal area (11.36 m$^2$ ha$^{-1}$) (Table S1). Generally, these forest stands had non-saline soils (EC = 0.13 to 0.99 dS m$^{-1}$) with slightly acidic (IPS = 6.84) to alkaline

**Table 1** Geospatial and soil physio-chemistry of three juniper forest sites in the study area.

| Variables | MPS | IPS | NPS |
|---|---|---|---|
| Latitude (°N) | 36.458 | 36.223 | 36.582 |
| Longitude (°E) | 72.657 | 72.495 | 72.848 |
| Altitude (m) | 2,624 | 2,607 | 2,668 |
| Slope angle (°) | 3.8 | 3.31 | 3.4 |
| Aspect (°) | 126 NW | 260 SW | 153 NW |
| pH(1:5) | 7.15 | 6.84 | 7.21 |
| EC (dS m$^{-1}$) | 0.998 | 0.135 | 0.39 |
| OM (%) | 0.68 | 0.31 | 0.98 |
| CaCO$_3$ (%) | 11.25 | 7.75 | 10.15 |
| Clay (%) | 17 | 20 | 18 |
| Sand (%) | 62 | 56 | 57 |
| Silt (%) | 21 | 24 | 25.5 |
| P (mg kg$^{-1}$) | 11.54 | 12.53 | 10.2 |
| K (mg kg$^{-1}$) | 171 | 133 | 157 |
| Conductivity (mm hr$^{-1}$) | 10.326 | 7.451 | 9.74 |
| Saturation (%) | 0.444 | 0.457 | 0.45 |
| Bulk density (g cm$^{-3}$) | 1.473 | 1.438 | 1.45 |
| Wilting point (kPa) | 0.117 | 0.128 | 0.12 |
| Water availability (%) | 0.102 | 0.109 | 0.11 |
| Field capacity (kPa) | 0.219 | 0.237 | 0.23 |

**Notes.**
MPS, Moderate Pruned Stand; IPS, Intensive Pruned Stand; NPS, Non-Pruned Stand.

nature (NPS & MPS $\leq$ 7.21). Organic matter ranged from 0.31 to 0.98% in the three forests, whereas CaCO$_3$ was high in MPS followed by NPS. Additionally, soil texture was predominantly sandy loam in all stands, with high K$^+$ content (133–171 mg kg$^{-1}$). Soil phosphorus was generally low, ranging from 10.2 to 12.53 mg kg$^{-1}$. Hydraulic properties such as saturation, bulk density, wilting point, water availability, and field capacity also varied across the three stands but did not differ significantly (Table 1).

## Impact of pruning on dendrometric and reproductive traits

In our study, we analyzed the effects of pruning on both *Juniperus semiglobosa* and *Juniperus excelsa* to identify the potential species-specific responses. However, there was a narrow margin in variation, and statistically non-significant differences between these two species in their response to pruning intensities were found. As a result, we collectively analyzed the data of both *Juniperus* species (junipers) for more concise and generalized findings. The density of juniper trees ranged from 137 to 156 (individuals ha$^{-1}$) with a basal area of 5.91 to 21.44 (m$^2$ ha$^{-1}$) and did not differ significantly between the experimental and control sites (Table 2). Similarly, the mean DBH and DRC ranged from 17 $\pm$ 0.75 to 20 $\pm$ 0.76 cm, and 23 $\pm$ 0.8 to 25 $\pm$ 1 cm respectively, with the highest values recorded at the moderately pruned site (MPS). Among the key structural attributes, the average tree volume (0.101 $\pm$ 0.012 m$^3$) was higher in moderately pruned sites (MPS) compared to intensively pruned sites (0.08 $\pm$ 0.008 m$^3$); however, the difference was not statistically

**Table 2** Key dendrometric parameters of junipers under various pruning treatments and the control site.

| | MPS | IPS | NPS | F-value | *p*-value |
|---|---|---|---|---|---|
| Density (individuals ha$^{-1}$) | 143 | 137 | 156 | N/A | N/A |
| BA (m$^2$ha$^{-1}$) | 15.77 | 5.91 | 21.44 | N/A | N/A |
| DBH (cm) | 20 ± 0.76$^a$ | 17 ± 0.75$^a$ | 19 ± 0.71$^a$ | 1.05 | 0.35 |
| DRC (cm) | 25 ± 1$^a$ | 23 ± 0.9$^a$ | 23 ± 0.8$^a$ | 1.65 | 0.19 |
| Volume (m$^3$) | 0.101 ± 0.012$^a$ | 0.08 ± 0.008$^a$ | 0.09 ± 0.007$^a$ | 1.2 | 0.29 |
| Age (Mean ± SE) | 33 ± 1.2$^b$ | 39 ± 1.6$^b$ | 66 ± 2$^a$ | 98 | **1.50E−27** |
| Age (min–max) | 17–55 | 24–52 | 40–169 | N/A | N/A |
| Height (m) | 5.98 ± 0.26$^a$ | 5.67 ± 0.23$^a$ | 4.8 ± 0.24$^b$ | 6 | **0.0024** |
| GR (cm year$^{-1}$) | 0.53 ± 0.02$^a$ | 0.34 ± 0.01$^b$ | 0.13 ± 0.004$^c$ | 304 | **4.55E−52** |
| Filled seeds (cone$^{-1}$) | 2.1 ± 0.2$^a$ | 2.1 ± 0.2$^a$ | 1.6 ± 0.13$^b$ | 28 | **0.0002** |
| Cones (numbers tree$^{-1}$) | 75 ± 7$^b$ | 70 ± 8$^b$ | 100 ± 4$^a$ | 7.96 | **0.0006** |

Notes.

DBH, Diameter at breast height; DRC, Diameter at root collar; GR, Growth rate.

*p* values in bold are significant. The superscript letters a, b, c indicate significant differences based on Tukey HSD test.

significant across the sites. The youngest recorded juniper tree was 17 years old, and the oldest was 55 years old at both moderately and intensively pruned sites. In contrast, older trees, ranging up to 169 years with an average age of 66 years were found in the non-pruned sites. Subsequently, a statistically significant difference in tree age ($f = 98$; $p = 1.50E\text{-}27$) was observed among the three experimental sites. Dendrometric parameters, including height and growth rate, along with reproductive traits, showed significant differences ($p \leq 0.001$) among the sites (Table 2). Specifically, the moderately pruned site had the tallest trees ($\mu = 5.98 \pm 0.26$ m), followed by the intensively pruned site, while the non-pruned sites contained shorter trees ($\mu = 4.8 \pm 0.24$ m). In contrast, seed viability was higher ($2.1 \pm 0.2$ seeds per cone) in both moderately and intensively pruned sites, although cone production varied significantly ($f = 7.96$; $p = 0.006$), being negatively affected by pruning, with the highest number of cones recorded in the non-pruned site ($100 \pm 4$ cones tree$^{-1}$). Additionally, our results revealed that both moderately and intensively pruned junipers had higher growth increments ($0.53 \pm 0.02$ and $0.34 \pm 0.01$ cm year$^{-1}$ respectively), compared to non-pruned site (Table 2).

## Impact of pruning on understory species composition and diversity

Pruning significantly impacts understory vegetation composition, density, and diversity. A total of 12 species were recorded in the understory stratum, predominated by shrubs (nine species) and a few herbs (three species) from 10 different families. The species composition in moderately and intensively pruned sites was largely similar, except for certain species, such as *Rosa webbiana*, *Berberis lycium*, *Sophora mollis*, *Lepidium sativum*, and *Taraxacum officinale*. Both the moderately pruned site (MPS) and intensively pruned site (IPS) hosted the highest number of species (nine species each). Understory densities across the three sites ranged from 5,411 to 7,714 individuals ha$^{-1}$, with the highest density in the MPS and the lowest in the NPS. Compared to the NPS, moderate and intensive pruning increased understory vegetation density by 11.45% and 7.77%, respectively. *Taraxacum*

**Table 3   Density (individuals ha$^{-1}$) of different understory species in the study sites.**

| Species | Habits | Local name | Family | MPS | IPS | NPS |
|---|---|---|---|---|---|---|
| *Rosa webbiana* Wall ex. Royle | Shrub | Throni | Rosaceae | 719 | 844 | 611 |
| *Artemisia scoparia* Waldst. & Kit | Herb | Drone | Asteraceae | 2,600 | 1,532 | * |
| *Berberis lycium* Royle | Shrub | Chownj | Berberidaceae | 323 | 478 | 1,733 |
| *Sophora mollis* Royle | Shrub | Beshu | Papilionaceae | * | 689 | 400 |
| *Daphne mucronata* Royle | Shrubs | Lovo mekin | Thymeleaceae | 400 | * | * |
| *Lepidium sativum* L. | Herb | Wah joshu | Brassicaceae | 548 | * | 756 |
| *Clematis orientalis* L. | Shrub | Chontruk | Ranunulaceae | * | 578 | * |
| *Haloxylon griffithii* (Moq) Bunge ex Boiss | Shrub | Pach | Chenopodiaceae | 1,067 | 319 | * |
| *Hippophae rhamnoides* L. | Shrub | Mirghinz | Eleaegnaceae | 623 | 756 | * |
| *Saccharum spontaneum* L. | Shrub | Shol | Poaceae | 234 | 400 | * |
| *Elaeagnus angustifolia* L. | Shrub | Sinjoor | Eleaegnaceae | 1,200 | 1,378 | * |
| *Taraxacum officinale* Weber | Herb | Phowu | Asteraceae | | * | 1,911 |
| Σ (summation of density) | | | | 7,714 | 6,974 | 5,411 |

**Notes.**
*Particular species were absent.

**Table 4   Shannon diversity index for understory vegetation in study sites.**

| | MPS | IPS | NPS |
|---|---|---|---|
| Shannon diversity index ($H$) | 1.93 | 2.07 | 1.43 |
| Evenness ($E$) | 0.88 | 0.944 | 0.89 |
| Total number of species richness ($S$) | 9 | 9 | 5 |
| Total number of individuals ($N$) | 7,714 | 6,974 | 5,411 |
| Average population size ($N^-$) | 857 | 775 | 1,080 |

*officinale*, a member of the Asteraceae family, was the most abundant herbaceous species, while *Berberis lycium* and *Rosa webbiana* were the most prominent shrubs across all sites (Table 3). Pruning also affected species diversity, with the Shannon–Weiner diversity index (H') being highest in the IPS (H' = 2.07), followed by the MPS (H' = 1.93). Species richness was similar in both pruned sites (nine species) but significantly lower in the NPS (five species). Species evenness (J') was comparable between the MPS (0.88) and NPS (0.89), but highest in the IPS (0.94) (Table 4).

## Impact of pruning on tree carbon biomass

Silvicultural treatments and their intensities had a significant impact on carbon biomass in juniper woodlands (Table 5). Moderate pruning proved to be more effective, increasing total living carbon (TLC) storage (4.95 Mg ha$^{-1}$), while higher pruning intensities led to a reduction in TLC stock (3.1 Mg ha$^{-1}$). Additionally, soil organic carbon (SOC) stock was adversely affected by more intense pruning. The highest SOC stock (nine Mg ha$^{-1}$) was observed in moderately pruned stands, whereas intensive pruning resulted in the lowest SOC stock (four Mg ha$^{-1}$). As a result, moderate pruning demonstrated the highest carbon sequestration potential (18.17 tons of $CO_2$ sequestered), while intensive pruning significantly reduced this capability to 11.38 tons of $CO_2$ sequestered.

**Table 5 Biomass and carbon stocks (Mg ha$^{-1}$) and CO$_2$ sequestration (ton CO$_2$ Seq.) of different pruned and control sites in the area.**

|  | SOC | AGB | AGBCS | BBCS | TLC | CO$_2$ Seq. |
|---|---|---|---|---|---|---|
| MPS | 9 | 8.26 | 4.13 | 0.82 | 4.95 | 18.17 |
| IPS | 4 | 5.18 | 2.59 | 0.51 | 3.1 | 11.38 |
| NPS | 6 | 6.55 | 3.77 | 0.75 | 4.52 | 16.59 |

**Notes.**

SOC, Soil organic carbon; AGB, Aboveground biomass; AGBCS, Aboveground biomass carbon stock; BBCS, Belowground biomass carbon stock; TLC, Total living carbon; CO$_2$ Seq, Carbon dioxide sequestration.

### Inter-correlation among dendrometric and reproductive characters

The linear relationships (Pearson, at 95% CI) among various dendrometric and reproductive traits under different silvicultural treatments are illustrated in Figs. 5 to 7. In addition, summary of Pearson correlation among the variables are presented in Table 6. In the non-pruned stands (NPS), tree DBH exhibited a strong positive correlation with DRC, and volume ($r > 0.92$, CI [0.85–0.91]), and moderate correlations with height ($r = 0.40$, CI [0.16–0.60]), age ($r = 0.33$, CI [0.07–0.54]), and tree growth rate ($r = 0.46$, CI [0.22, 0.46]). Similarly, DRC followed a comparable correlation pattern, except for height ($r = 0.53$, CI [0.31–0.70]). However, no significant correlation was found between reproductive traits and dendrometric variables. Height was significantly correlated with age ($r = 0.47$, CI [0.24–0.65]) and volume ($r = 0.63$, CI [0.44–0.77]), suggesting that age may serve as a good predictor for both these traits. In the moderately pruned stand (MPS), the pruning effects became evident as DBH showed slightly lower correlations with DRC ($r = 0.80$, CI [0.67–0.74]) and volume ($r = 0.88$, CI [0.71–0.74]) compared to NPS, but a higher correlation with height ($r = 0.68$, CI [0.54–0.59]), indicating enhanced height increment due to moderate pruning. Interestingly, in MPS, DBH had a non-significant weak correlation with growth rate, while DRC maintained a similar relationship with height. A notable shift was observed in growth rate, which was negatively impacted by age ($r = -0.53$ CI [−0.48–0.15]), a pattern absent in NPS. In the intensely pruned stand (IPS), DBH correlations weakened further with DRC ($r = 0.57$, CI [0.52–0.58]) and height ($r = 0.48$, CI [0.45–0.53]), reflecting the impact of intense pruning on these parameters. The reproductive traits, including seed and cone production, displayed a similar correlation trend as in NPS and MPS. Moreover, the growth rate in IPS was negatively correlated with age ($r = -0.30$, CI [−0.45 to −0.33]) and height ($r = -0.23$, CI [−0.23 to −0.02]), mirroring the pattern in MPS. The negative correlation between age and growth rate became more pronounced, while age and height maintained a positive association.

## DISCUSSIONS

Junipers are long-lived and characterized by slow growth patterns, found in the high mountainous environments of northern Pakistan (*Esper, 2000*; *Ahmed et al., 2011*), and neighbouring countries (*Bakhtiyorov et al., 2023*; *Opała-Owczarek et al., 2023*). Studies on the pruning of juniper trees especially of *J. excelsa* and *J. semiglobosa* were not documented, even though these silvicultural techniques may have an impact on their health and vitality.

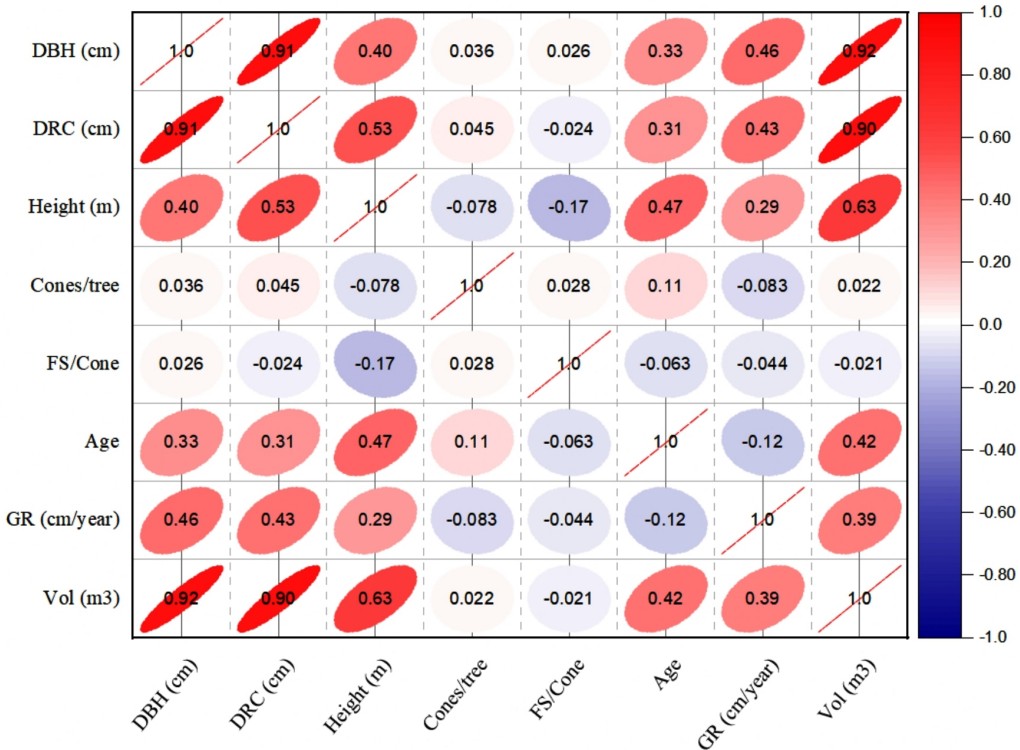

**Figure 5  Pearson correlations among the reproductive traits and dendrometric factors in non-pruned stand (NPS).** Keys: DBH, Diameter at breast height; DRC, Diameter at root collar; FS, filled seeds; GR, growth rates; Vol, stem-volume.

Likewise, improper and non-scientific silvicultural treatments can disrupt their natural developmental processes, resulting in ecological imbalances, reduced productivity, and in extreme cases species mortality. During the current study, we evaluated traditional pruning treatments and their impact on forest structural parameters, carbon mitigating potential, soil physio-chemical properties, and their impacts on understory vegetation dynamics. Our study revealed that pruning in junipers plays a significant role in regulating forest ecological metrics and is found to be an important forest management practice. Individuals that underwent pruning showed knot-free wood, improved height, better diameter growth, and enhanced reproductive traits. These effects might be because pruning may have an efficient effect on physiological processes occurring in plants. Many researchers have agreed that pruning has a positive impact on photosynthesis in plants (*Ares & Brauer, 2005*; *Suchocka et al., 2021*), also improves leaf nitrogen content which has a significant role in yields (*Maurin & DesRochers, 2013*) and delay leaf senescence (*Suchocka et al., 2021*). It is important to note that other factors, such as inter-specific competition, tree age, and microenvironment (*e.g.*, light and soil nutrient availability) could also influence the structural parameters (*Nabeshima, Kubo & Hiura, 2010*; *Stephenson et al., 2014*). In our study, the forests were composed of a few species and their contribution in terms of importance values and density was low, suggesting that competition may not significantly
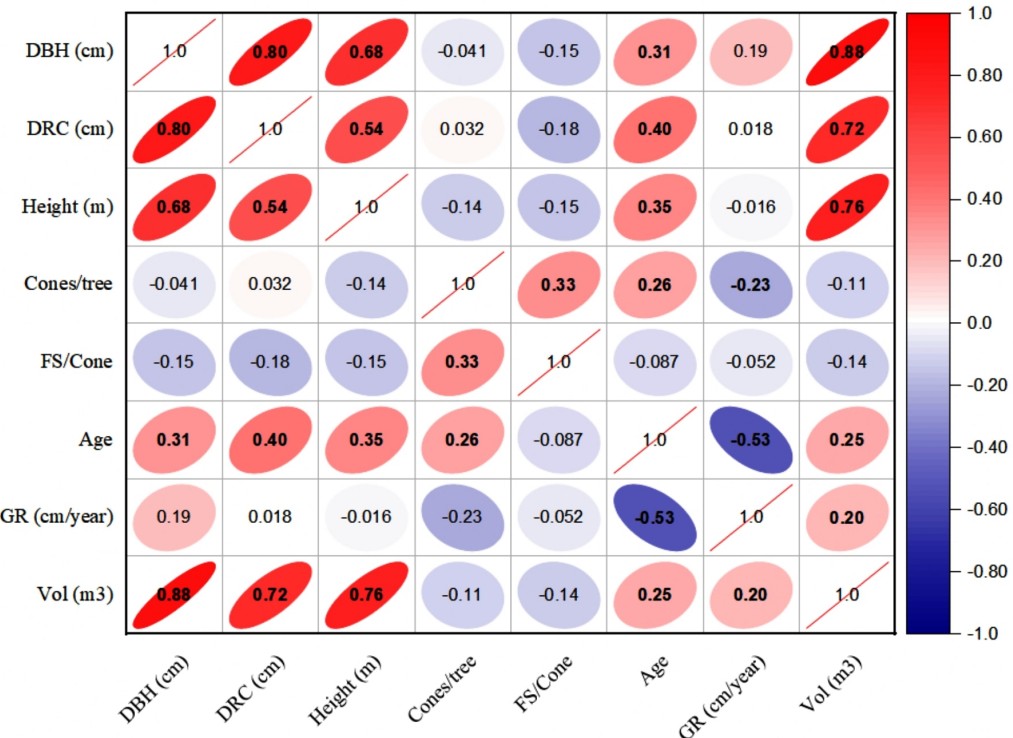

**Figure 6** **Pearson correlations among the reproductive traits and dendrometric factors in Moderate Pruned Stand (MPS).** For variables, see the key in Fig. 5.

affect tree diameter. Similarly, the microenvironment had minimal impact due to the proximity of the stands and the homogeneity of environmental conditions.

Pruning treatment significantly influenced dendrometric characters in the study sites. Our results revealed that moderate pruning (30–40%) was beneficial to regulate diameter growth with non-significant variation compared to non-pruned and intensively pruned sites, while maximum height was recorded both in MPS and IPS with statistically significant variation compared to NPS. Conversely, pruning with higher intensity (> 60%) negatively influenced diameter, these findings aligned with previous studies (*Huang et al., 2023*; *Li et al., 2020*; *Hevia, Álvarez-González & Majada, 2016*; *Springmann, Rogers & Spiecker, 2011*). Moreover, intensive pruning can also increase the risk of dieback, particularly in mature trees (*Suchocka et al., 2021*). *Dykstra & Monserud (2007)* reported that pruning produced larger trees with higher DBH, similarly *Forrester & Baker (2012)* also reported improved DBH size after pruning treatment. Furthermore, *Strong & Erdmann (2000)* observed that both thinning and pruning treatments were collectively found to be the influential factors in boosting diameter growth. Pruning is likely to have resulted in a reduction in tree leaf area and transpiration rates accompanied by improvements in light and water use efficiency and the rate of $CO_2$ assimilation per unit leaf area (*Forrester et al., 2012*; *Lisboa et al., 2014*). Similarly, study show that pruning improves light interception in trees by enhancing light penetration and reducing self-shading. After one growing season, pruned

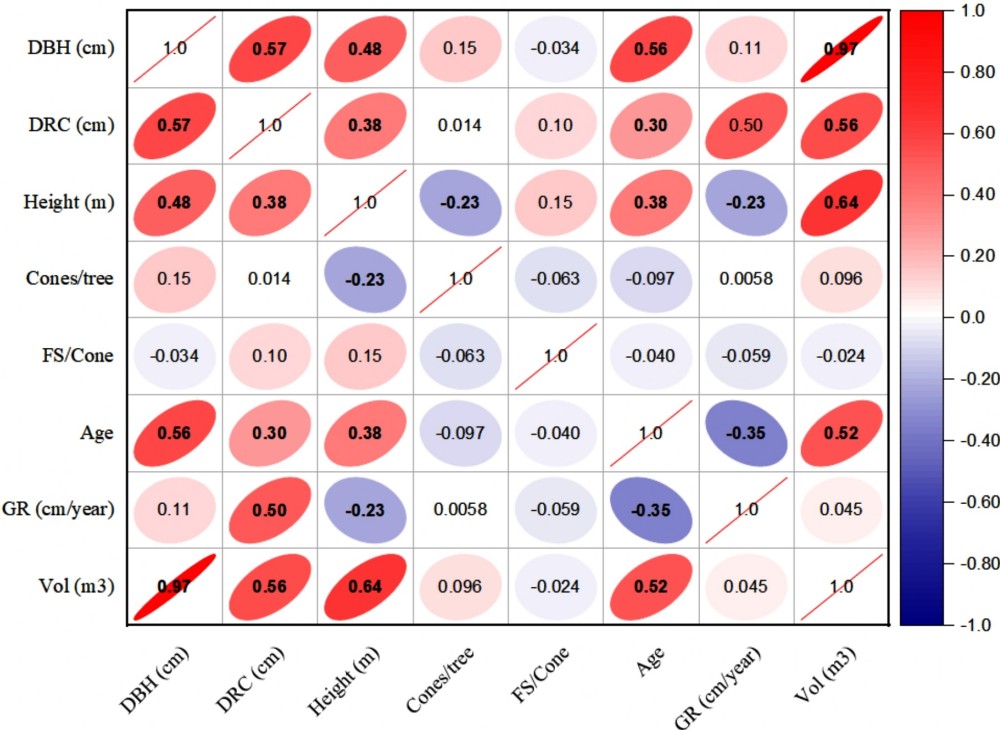

**Figure 7  Pearson correlations among the reproductive traits and dendrometric factors in intensive pruned stand (IPS).** For variables, see the key in Fig. 5.

trees recover their light interception while achieving a more open, cup-shaped structure. Consequently, affects more efficient light usage, particularly in the deeper parts of the crown, ultimately benefiting tree growth and health (*Tosto et al., 2022*). In *Prosopis* species pruning has been practiced to improve the shape (short-term) and increase stem diameter (long-term), as *Prosopis* species are pruned worldwide (*Alvarez et al., 2013*). In addition, pruned trees quickly compensated for the light interception, which was closely linked to increased leaf-flushing activity (*Tosto et al., 2022*). This response was commonly observed in trees subjected to pruning, such as cocoa (*Leiva-Rojas et al., 2019*), mango (*Persello et al., 2019*), and apple (*Fumey et al., 2011*). Instead, some previous studies have also suggested the negative impact of pruning on tree height and diameter growth (*Pinkard & Neilsen, 2003*; *Zeng, 2003*). Our investigation revealed that juniper stands subjected to pruning exhibited growth rates higher than those observed at the control site. Interestingly, our analysis indicated that intensive pruning led to slightly lower growth rates compared to moderate pruning, but the growth rate was still higher than non-pruned individuals. Consequently, higher growth rates directly influenced the diameter growth resulting high quality wood. These findings align well with studies conducted by *Víquez & Pérez (2005)* and *Danilović et al. (2022)*. It is reported that pruning severity exceeding 60% in poplar clone 'I-214' at the age of 7 years had a negative impact on growth rates (*Keller, 1979*).

**Table 6 Summary of Pearson correlation coefficients among the variables, with 95% confidence intervals (CIs) calculated using Fisher's z-transformation.**

| Variable 1 | Variable 2 | NPS | | | MPS | | | IPS | | |
|---|---|---|---|---|---|---|---|---|---|---|
| | | r-value | L-CI | U-CI | r-value | L-CI | U-CI | r-value | L-CI | U-CI |
| DBH | DRC | 0.91 | 0.85 | 0.95 | 0.80 | 0.67 | 0.74 | 0.57 | 0.52 | 0.58 |
| DBH | Height | 0.40 | 0.16 | 0.60 | 0.68 | 0.54 | 0.59 | 0.48 | 0.45 | 0.53 |
| DBH | Cones | 0.04 | −0.23 | 0.30 | −0.03 | −0.03 | 0.29 | 0.27 | −0.03 | 0.26 |
| DBH | FS | −0.04 | −0.30 | 0.23 | 0.39 | 0.23 | 0.37 | 0.12 | 0.12 | 0.36 |
| DBH | Age | 0.33 | 0.07 | 0.54 | 0.31 | 0.30 | 0.50 | 0.56 | 0.29 | 0.51 |
| DBH | GR | 0.46 | 0.22 | 0.64 | 0.19 | 0.18 | 0.57 | 0.11 | 0.11 | 0.18 |
| DBH | Volume | 0.92 | *0.86* | *0.95* | 0.88 | 0.71 | 0.74 | 0.97 | 0.61 | 0.75 |
| DRC | Height | 0.53 | 0.31 | 0.70 | 0.54 | 0.49 | 0.60 | 0.38 | 0.36 | 0.46 |
| DRC | Cones | 0.05 | −0.22 | 0.31 | −0.07 | −0.07 | 0.30 | 0.07 | −0.07 | 0.07 |
| DRC | FS | −0.04 | −0.30 | 0.23 | −0.42 | −0.39 | 0.23 | −0.36 | −0.38 | −0.35 |
| DRC | Age | 0.31 | 0.05 | 0.53 | 0.40 | 0.38 | 0.49 | 0.30 | 0.29 | 0.37 |
| DRC | GR | 0.43 | 0.18 | 0.62 | 0.02 | 0.02 | 0.55 | 0.50 | 0.02 | 0.46 |
| DRC | Volume | 0.90 | 0.83 | 0.94 | 0.72 | 0.61 | 0.74 | 0.56 | 0.51 | 0.55 |
| Height | Cones | −0.08 | −0.34 | 0.19 | −0.10 | −0.10 | 0.19 | −0.11 | −0.11 | −0.10 |
| Height | FS | −0.14 | −0.39 | 0.13 | 0.00 | 0.00 | 0.13 | 0.48 | 0.00 | 0.44 |
| Height | Age | 0.47 | 0.24 | 0.65 | 0.35 | 0.34 | 0.57 | 0.38 | 0.32 | 0.37 |
| Height | GR | 0.29 | 0.02 | 0.51 | −0.02 | −0.02 | 0.47 | −0.23 | −0.23 | −0.02 |
| Height | Volume | 0.63 | 0.44 | 0.77 | 0.76 | 0.64 | 0.65 | 0.64 | 0.57 | 0.57 |
| Cones | FS | 0.25 | −0.02 | 0.48 | −0.34 | −0.32 | 0.45 | −0.30 | −0.31 | −0.29 |
| Cones | Age | 0.11 | −0.16 | 0.37 | 0.09 | 0.09 | 0.35 | 0.08 | 0.08 | 0.09 |
| Cones | GR | −0.08 | −0.34 | 0.19 | −0.02 | −0.02 | 0.18 | −0.05 | −0.05 | −0.02 |
| Cones | Volume | 0.02 | −0.25 | 0.29 | −0.06 | −0.06 | 0.28 | 0.22 | −0.06 | 0.22 |
| FS | Age | 0.03 | −0.24 | 0.29 | −0.06 | −0.06 | 0.28 | 0.51 | −0.06 | 0.47 |
| FS | GR | −0.14 | −0.39 | 0.13 | 0.29 | 0.13 | 0.28 | −0.54 | −0.49 | 0.27 |
| FS | Volume | −0.05 | −0.31 | 0.22 | 0.29 | 0.22 | 0.28 | 0.27 | 0.26 | 0.27 |
| Age | GR | −0.12 | −0.38 | 0.15 | −0.53 | −0.48 | 0.15 | −0.35 | −0.45 | −0.33 |
| Age | Volume | 0.42 | 0.17 | 0.62 | 0.25 | 0.25 | 0.55 | 0.52 | 0.24 | 0.48 |
| GR | Volume | 0.39 | 0.14 | 0.60 | 0.20 | 0.20 | 0.53 | 0.05 | 0.05 | 0.20 |

**Notes.**

L-CI, Lower confidence interval; U-CI, Upper confidence interval, for other variables, see the key in Fig. 5.

Additionally, there were higher densities of understory vegetation in both pruned sites compared to non-pruned stand, potentially due to the availability of proper light. On the other hand, it is reported that pruning might negatively affect the sciophytes (*Halpern et al., 2012*; *Roberts et al., 2016*). Reduction in canopy size of forest may alter understory species composition, richness, and densities by influencing light and temperature (*Berrill et al., 2017*; *Zhao et al., 2021*). Furthermore, the biomass of understory vegetation increases with pruning intensity likely due to the accumulation of photosynthetic products. Canopy pruning enhances the intensity of light penetration in the understory, thereby increasing photo-synthetically active radiation which likely promotes the growth of the vegetation found on the forest floor (*Ares, Louis & Brauer, 2003*; *Brockerhoff et al., 2003*). In the current

study, we observed an increase in the Shannon diversity index of understory vegetation within pruning treatments, especially in high-intensity pruning site. However, species richness remained unaffected by pruning intensity and was consistent both in moderately and intensively pruned sites, but low species richness was recorded in NPS.

Our results show that both AGBC and BBC stocks were found to be higher in MPS, which likely due to the significant role of pruning in growth rate and diameter girth, which is a key contributor to carbon biomass. However, high-intensity pruning had a negative impact on carbon storage and $CO_2$ sequestration as obvious from the results, suggesting that normal pruning within a certain range could be effective in climate change mitigation and achieving clear bowel in a limited time. Similar findings were reported in a previous study conducted by *Medhurst et al. (2006)*, where pruning significantly affects forest structure and improves carbon sequestration. They reported an increase of up to 50% in photosynthetic capacity between 2 and 6 weeks after pruning in *Acacia melanoxylon*, depending on the crown region. Additionally, carbon sequestration was notably higher in upper crown pruning treatments by 33%, and in middle crown pruning treatments by 62% compared to non-pruned controls (*Muhammad et al., 2023*; *Medhurst et al., 2006*). Our results have also been supported by *Muhammad et al. (2023)* and *Neilsen & Pinkard (2003)* stated that heavy pruning can reduce photosynthetic capacity and tree growth, thereby decreasing the forest's ability to sequester carbon. In previous studies, it has been reported that *Eucalyptus nitens* stands accumulate substantial (85–90%) biomass in stem wood and bark, where bark contributed 9–12% of AGB compared to stem wood (*Muñoz et al., 2008*; *Xue et al., 2023*; *Li et al., 2023*). However, *Liu et al. (2019)* reported contradictory findings suggesting that thinning and pruning did not yield a sustained increase in the overall net carbon stock. Furthermore, soil organic carbon stocks (SOC) were also higher in the MPS, and lower SOC was found in intensively pruned site. *Gómez-Muñoz et al. (2016)* suggested that pruning can be good for increasing the amount of carbon stored in the soil because the leftover tree parts from pruning can slowly release organic matter into the soil over time. However, these results are contrary to existing literature (*e.g.*, *Alcorn et al., 2008*; *Bussi et al., 2010*; *Chen et al., 2016*; *Forrester et al., 2012*) and might be attributed to differences in studied species, site conditions, and method of pruning by the local peoples. Moreover, various silvicultural treatments (like thinning and pruning) exert an influence on stand structure and density, thereby directly impacting the dynamic balance of living biomass and quantities of dead wood, ultimately leading to alterations in litter-fall inputs, therefore influencing soil carbon storage, as documented by *Roig et al. (2005)* and *Blanco, Imbert & Castillo (2006)*.

## CONCLUSION AND RECOMMENDATIONS

The impact of pruning practices on *Juniperus* species in the eastern Hindu Kush region provides valuable insights into the ecological implications of different pruning intensities. Results indicate that moderate pruning within the 30–40% range has a positive influence, whereas pruning exceeding 60% may have adverse effects on key dendrometric. This implies a delicate balance in pruning practices, where moderation is crucial to avoid

negative impacts on the tree metrics. Furthermore, the study highlights the importance of pruning intensity in influencing understory vegetation, soil nutrient dynamics, and the carbon storage capacity of junipers. The positive impacts of moderate-pruning on living carbon biomass and soil organic carbon density, highlights the potential benefits of carefully managed-silvicultural practices. Likewise, we suggest adopting a balance pruning of juniper trees for improving forest productivity. This strategy could enhance ecosystem services, including carbon sequestration, particularly in the face of climate change, and securing stable economic benefits for local inhabitants reliant on these ecosystems.

## ACKNOWLEDGEMENTS

All the authors are thankful to the Pakistan Meteorological Department (PMD) for sharing the climate data of the Chitral meteorological station. The officials of the Agricultural Research Station (ARI) Chitral are acknowledged for soil physiochemical analysis.

### Funding

This research was supported by the King Saud University Riyadh Saudi Arabia (Grant No. RSP2023R374). There was no additional external funding received for this study. The funders had no role in study design, data collection and analysis, decision to publish, or preparation of the manuscript.

### Grant Disclosures

The following grant information was disclosed by the authors:
The King Saud University Riyadh Saudi Arabia: RSP2023R374.

### Competing Interests

The authors declare there are no competing interests.

### Author Contributions

- Nasir Ud Din conceived and designed the experiments, performed the experiments, analyzed the data, prepared figures and/or tables, and approved the final draft.
- Nasrullah Khan conceived and designed the experiments, performed the experiments, analyzed the data, prepared figures and/or tables, authored or reviewed drafts of the article, and approved the final draft.
- Rafi Ullah performed the experiments, analyzed the data, authored or reviewed drafts of the article, and approved the final draft.
- Mohammad K. Okla conceived and designed the experiments, analyzed the data, authored or reviewed drafts of the article, and approved the final draft.
- Mostafa A. Abdel-Maksoud performed the experiments, authored or reviewed drafts of the article, and approved the final draft.
- Ibrahim A. Saleh performed the experiments, analyzed the data, authored or reviewed drafts of the article, and approved the final draft.

- Hashem A. Abu-Harirah analyzed the data, authored or reviewed drafts of the article, and approved the final draft.
- Tareq Nayef AlRamadneh analyzed the data, authored or reviewed drafts of the article, and approved the final draft.

## Data Availability

The raw data is available in the Supplementary File.

## Supplemental Information

Supplemental information for this article can be found online at http://dx.doi.org/10.7717/peerj.19184#supplemental-information.

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
