# Peer review of "Does pruning affect the structural and ecological productivity of Juniper woodlands in the eastern Hindu Kush?"

_PeerJ, doi:10.7717/peerj.19184_

## Round 0.1 · original submission · Major Revisions

Please consider addressing all comments of the reviewers. Further to their comments, (1) what are the confidence intervals for your tests? There should be a table indicating these quantities. (2) The Pearson correlation coefficients also should have confidence intervals calculated - you can do that over the z-transformation. Please include this in working with your figures, update them to include the CI and discuss all results in light of confidence intervals. (3) discuss critically the effect of the sample number, (4) the statistical section, especially the methodology needs to be reworked/ overhauled to become clearer and significantly improved; please check all (statistical) model requirements with the data at hand (eg do extra distribution tests, and illustrate distributions/assumptions). (5) Make sure that the broader scope of this research is better addressed in the Discussion section - what are the implications for other forests, can you discuss this using other literature and how do you link this to the information provided in the introduction. (6) The manuscript is in parts quite repetitive - removal of repetition and making the manuscript more concise is paramount before acceptance, try to make the text "more precise" AND "more concise". Thanks

Reviewer 1 ·

Basic reporting

no comment

Experimental design

no comment

Validity of the findings

no comment

Additional comments

Dear authors,

Your manuscript "Does pruning affect the structural and ecological productivity of Juniper woodlands in eastern Hindukush" has now been assessed. I have performed a preliminary analysis of your paper. Here is some comments, addressed all the comments and modify the paper grammatically as well.
• The title could benefit from improved clarity in phrasing. For example, “Does pruning affect the structural and ecological productivity of Juniper woodlands in the eastern Hindukush?” This revised phrasing improves readability and clearly distinguishes “structural and ecological productivity” as aspects affected by pruning.
• Abstract, line-18, Juniperus species or genus if this is species then mention the species information
• Line 18-scree slopes? What does it mean, confusing
• Too much lengthy abstract, reduce upto 250 words.
• Revise the keywords, donot use 2 words and nor consider 2 words as single keyword
• Novelty is absent in the paper, add novelty statement at the end of introduction
• Line 112-113, Quercus incana, Q. dilatata, Q. baloot, Juglans regia, Tamarix species, and Betula utilis?? Irregularity, some plant names are written in full form some are in abbreviated form, while again there is unknown species of Tamarix? Mention the particular specie name.
• Modify the results part by English native speaker and provide certificate
• The same is for overall section, modify the paper English, remove grammatical mistakes, remove extra spacing errors, spelling, replace comma by full stop etc, there is a lot of things which needs improving
• Compare your own results with latest literature review (up to date from 2020-2025)
• Revise conclusion
• Replace figure 3, draw new one, having more than 400 dpi
• Avoid the self-citation
1. Khan N, Ahmed M, Shaukat S. 2013. Climatic signal in tree-ring chronologies of Cedrus deodara from Chitral Hindu Kush Range of Pakistan. Geochronometria 40:195–207.
2. Khan N, Ali F, Ali K, Shaukat S. 2015. Composition, structure and regeneration dynamics of
Olea ferruginea Royle forests from Hindukush range of Pakistan. Journal of Mountain
Science 12:647–658
3. Khan N, Nguyen HT, Galelli S, Cherubini P. 2022. Increasing drought risks over the past four
centuries amidst projected flood intensification in the Kabul River Basin (Afghanistan
and Pakistan)—Evidence from tree rings. Geophysical Research Letters 49(24).
4. Ahmed M, Palmer J, Khan N, Wahab M, Fenwick P, Esper J, Cook ED. 2011. The dendroclimatic potential of conifers from northern Pakistan. Dendrochronology 29(2):77–88.
5. Ali F, Khan N, Ali K, Amin M, Khan MEH, Jones DA. 2023. Assessment of variability in
nutritional quality of wild edible fruit of Monotheca buxifolia (Falc.) A. DC. along the
altitudinal gradient in Pakistan. Saudi Journal of Biological Sciences 30(1):103489
Remove your own references from the paper, this is unethical


Kind regards,

Annotated reviews are not available for download in order to protect the identity of reviewers who chose to remain anonymous.

Reviewer 2 ·

Basic reporting

This study explores an intriguing topic: the management of natural woodlands for forest products. While the study is well-structured, there are key aspects that require attention before it can be considered for acceptance.

Firstly, the English language needs improvement to ensure clarity and precision. Secondly, the hypothesis presented in lines 89–90—"The current study therefore hypothesized that pruning has beneficial impacts on the juniper forests"—is not well integrated into the preceding context. While it is reasonable to assume that removing stems could enhance the growth of the remaining ones, the hypothesis appears disconnected from the rationale provided earlier in the manuscript.

To strengthen this section, the authors should establish a coherent and logical progression leading to the hypothesis. This includes clearly explaining how pruning aligns with the ecological or silvicultural principles discussed earlier, supported by relevant literature or preliminary observations. Doing so will provide a solid foundation for the study's objectives and improve its overall scientific rigor.

Experimental design

The authors refer to Juniper species in the manuscript. However, it is unclear how many Juniper species were analyzed in this study. Clarifying this point is crucial, as the ecological and physiological responses to pruning might vary between species.

To enhance the quality and depth of the study, the authors should consider conducting separate analyses for the different Juniper species, if multiple species were indeed studied. This approach would provide more specific insights and strengthen the ecological implications of the findings.

Validity of the findings

While the findings are locally relevant, they contribute to the already extensive literature on this particular topic. To enhance the novelty and broader applicability of the study, the authors could highlight unique aspects of their research setting or methodology that distinguish it from previous studies. Emphasizing these points would better demonstrate how this study advances understanding within the field.

---

## Round 0.2 · Minor Revisions

The authors are advised to apply the following corrections/ address additional concerns to improve the manuscript:

- Please explain 'ecological productivity'.
- explain better the formulae. E.g. Equation 2: "CF is the conversion factor for rock fragments, which was not applicable in our case", i.e. CF is set to 1, or omitted?
- Double check the units; e.g. (Mg could be confused with magnesium; it should be mg.
Also, check the grammar of your manuscript and try to make it more concise. E.g. "growing either as mono-specific stands or co-occur in.." must be "...co-occuring". E.g. in the conclusion section, you can eliminate "Our study," "our findings", etc. just present the facts. Try to shorten throughout the text."

Reviewer 1 ·

Basic reporting

NA

Experimental design

NA

Validity of the findings

NA

Additional comments

NA

Reviewer 2 ·

Basic reporting

The manuscript has been improved and all my comments addressed. In my opinion, it is now acceptable in its present form for its pubblication

Experimental design

The manuscript has been improved and all my comments addressed. In my opinion, it is now acceptable in its present form for its pubblication

Validity of the findings

The manuscript has been improved and all my comments addressed. In my opinion, it is now acceptable in its present form for its pubblication

---

## Round 0.3 · accepted · Accept

The paper is ready for publication.

The Section Editor noted:

> I suppose Mg in equation 3 is mega-gramms? In this case, I suggest writing 10E6 g. Otherwise it could be confused with 'Magnesium.